# Bridge Deformation Analysis Using Time-Differenced Carrier-Phase Technique

**María Jesús Jiménez-Martínez** [1,*] , **Nieves Quesada-Olmo** [1] , **José Julio Zancajo-Jimeno** [2] **and Teresa Mostaza-Pérez** [2]

1 Department of Cartographic Engineering, Geodesy, and Photogrammetry, Universitat Politècnica de València, 46005 València, Spain
2 Department of Cartographic and Land Engineering, Universidad de Salamanca, 37008 Salamanca, Spain
* Correspondence: mjjimenez@cgf.upv.es; Tel.: +34-6447-716-877

**Abstract:** Historically, monitoring possible deformations in suspension bridges has been a crucial issue for structural engineers. Therefore, to understand and calibrate models of the "load-structure-response", it is essential to implement suspension bridge monitoring programs. In this work, due to increasing GNSS technology development, we study the movement of a long-span bridge structure using differenced carrier phases in adjacent epochs. Many measurement errors can be decreased by a single difference between consecutive epochs, especially from receivers operating at 10 Hz. Another advantage is not requiring two receivers to observe simultaneously. In assessing the results obtained, to avoid unexpected large errors, the outlier and cycle-slip exclusion are indispensable. The final goal of this paper is to obtain the relative positioning and associated standard deviations of a stand-alone geodetic receiver. Short-term movements generated by traffic, tidal current, wind, or earthquakes must be recoverable deformations, as evidenced by the vertical displacement graphs obtained through this approach. For comparison studies, three geodetic receivers were positioned on the Assut de l'Or Bridge in València, Spain. The associated standard deviation for the north, east, and vertical positioning values was approximately 0.01 m.

**Keywords:** bridge monitoring; GNSS; time-differenced carrier phases; cycle-slip detection; least squares

## 1. Introduction

The socioeconomic impact represented by the construction of important civil engineering infrastructures, especially large-scale bridges, is indisputable. These structures, which are part of a country's heritage, play a crucial role in various aspects of modern life and are essential to promote economic growth and the connection of communication routes, in addition to facilitating the flow of mobility.

However, extreme loads, aging, and increased traffic volume can affect a bridge's serviceability, and many of these structures will soon be nearing the end of their original design life. Even so, in general, civil structures can continue to be used beyond their useful design life, and it is necessary for them to function safely. For this reason, damage detection techniques are developed and implemented since these structures cannot normally be replaced because of their significant economic cost. It is increasingly common for civil engineers to develop riskier and more efficient and profitable structural designs, in relation to design and novel engineering systems, by using the latest materials for which long-term response and degradation processes need to be better understood. Given these circumstances, in new systems, it is required that the appearance of damage be detectable as soon as possible. For this reason, the development and implementation of a structural health monitoring (SHM) system are considered exceedingly important steps to evaluate the performance of bridges in an effort to prevent failures, which can have serious economic and life safety consequences [1].

The first known examples of regular bridge monitoring, as conducted by Carder [2], were carried out on the Bay Bridge and Golden Gate Bridge in San Francisco in a project that encompassed several measurement periods of the different components while construction of these bridges was underway to ascertain the possible consequences and dynamic behaviour during an earthquake [3]. From the 1900s onwards, the monitoring of suspension bridges developed great interest, increasing the potential of this technique. Highlights include the Humber Bridge, in which approximately 70 instruments were used to measure the temperature, acceleration, displacement, and rotation of the bridge structure together with the speed and direction of the wind. The principal aim was to validate the mathematical models that had been developed of the responses to the wind in long-span bridges and to define the relationships between the parameters of the load and the response of the structure [4].

On the one hand, accelerometers used as vibration frequency monitoring tools require a double integration to obtain the displacement values, and they also need a trend element that is generated when the integration process is carried out, as they are not able to measure the quasistatic displacement over a long period of time [5]. On the other hand, one of the keys to evaluating the drift and stress parameters of the structures is the relative displacements, which are difficult to measure directly. Celebi et al. [6] defined structural monitoring systems using GPS technology as an alternative method to measure such relative displacements.

Due to its attractive advantages compared to usual sensors used in traditional bridge monitoring, the Global Navigation Satellite System (GNSS) is presently fully implemented in high-precision structural health monitoring for dams, bridges, other civil engineering infrastructure, and high-rise buildings. Notable in this case is the University of Nottingham, United Kingdom, which has been conducting a study of the structural health monitoring of bridges with GNSS technologies since the year 2000. Given the practice and research of the dynamic responses of bridges, Meng's thesis focused attention on the control of the dynamic deformation of the bridge. "Zero baseline (ZBL) and short baseline (SBL) tests were conducted to evaluate the performance of three types of Leica GPS receivers at a 10 Hz sampling rate. The statistical characteristics of positioning solutions and the achievable accuracy of each receiver type were analysed. The results were then employed to design optimal filters for various GPS error suppressions". In Meng's thesis, if ambiguities are fixed, and appropriate filtering techniques are enforced on the positioning solutions, it was presented that there are possibilities of achieving millimetre-level monitoring in the deformation of bridges [7]. The Millennium Bridge on the Thames in London, the Forth Road Bridge in Scotland, and the Wilford Suspension Bridge in Nottingham have applied the results obtained at the University of Nottingham [5]. Peppa et al. used GPS signal-to-noise measurements to measure bridge vibrations [8]. A specific GNSS and Earth observation SHM system called GeoSHM was elaborated by Meng et al. in recent years and has been implemented on various bridges in China, such as the Erqi Yangtze River Bridge and the Zhixi Yangtze River Bridges, as well as the Forth Road Bridge in Scotland [9].

Within the same field of structural health monitoring, interesting studies related to the influence of thermal action and displacement in cable-stayed bridges are being conducted in parallel [10,11]

Multi-GNSS has been used to enhance the precision of position estimations for structural movement detection due to the increase in global navigation satellite systems. It allows for the use of the kinematic PPP technique for dynamic deflection measurement with 1 s range satellite clock correction. Tang et al. applied it to the towers and suspension cables of Severn Bridge in the UK. Their investigation shows that tiny movements can be perceived with high-rate satellite clock corrections, which is impossible when using the 300 s satellite clock products from the IGS. According to this, the authors offer a different approach for the precise structural monitoring with PPP when the Double-Difference method (DD) cannot be employed [12].

Xin et al. [13], although they adopted the sampling period of 10 s, used a 9000-sample collection measured via GNSS of the mid-span deformation of the Caijia Jialing River Bridge in China to explore the performance of the Kalman-ARIMA-GARCH model proposed. This has better performance in predicting the deformation compared with the traditional linear ARIMA model.

After that, other investigations showed that the wavelet transform (WT) can be employed to perform frequency domain analysis, smooth the GPS measurements, and detect damage to structures. By exploring both analytical and GPS measurements, the authors Hussan et al. [14] conducted a study on the actions of the Incheon Bridge located in Korea under the effect of loads. The data of GPS observation have no noise, and the movements have been mined using multi-filtering approaches. The dominating energy was assessed by applying the wavelet decomposition method, which evidences the dynamic of the structure.

In the same way, Kaloop et al. [15] evaluated the full behaviour of the Incheon bridge by utilizing the RTK-GPS measurements for short-period monitoring. They concluded that to remove the noises of GPS signals, a convenient tool capable of it is the wavelet-transform method. The errors and parameters of the autoregressive-moving average (ARMA) model can evaluate the behaviour, rigidity, and durability of the bridge.

With the GNSS dynamic monitoring data of the Sutong Yangtze River Bridge, as the study object, a method of analysis of time–frequency founded on the combination of wavelet threshold denoising and Hilbert–Huang transform (HHT) was advanced. With this method, Wang et al. [16] analysed the dynamic characteristics of the bridge, and the authors identified the natural vibration of the structure.

Xi et al. proposed a processing method that integrates all GNSS constellations available at the time of observation (multi-GNSS), which allows for the evaluation of the performance of multi-GNSS resolution with high-elevation cut-offs [17].

Stiros, Moschas, and Triantafyllidis [18] obtained experimental and statistical evidence. It was discovered that for a high signal-to-noise ratio (SNR), deviation differences of 0.3 Hz from the correct value could be found in the spectral peaks in acceleration and displacement. For a low signal-to-noise ratio, only the acceleration spectra match the actual frequency. When various excitation effects exist (traffic, wind, etc.), they contribute a large number of peaks in a diverse range of frequencies.

In connection with previous investigations by the same author, Stirios [19] examined a characteristic case of painstaking monitoring of a reinforced concrete road bridge with huge dynamic deflections and explained that these deviations are generated due to a double process caused by the great reflective surfaces of moving vehicles near the antenna.

The use of low-cost GNSS devices for structural health monitoring is beginning to be developed in depth. Manzini et al. [20] used a data set observed over two weeks with a network of low-cost GNSS receivers deployed on a suspension bridge to validate on-site performance. The authors highlight in their conclusions the future potential of low-cost devices for structural health-monitoring developments.

Xue, Psimoulis, Horsfall et al. [21] assessed the performance of the response in monitoring the bridge with low-cost GNSS receivers and the estimation of the response amplitude and modal frequency. The study was conducted at the Wilford Suspension Bridge in the UK. In addition, regarding the improvement of the ambiguity correction rate, the benefit of the multi-GNSS solution in low-cost GNSS receivers was confirmed due to the deviation of a few mm in the estimation of the response of bridge amplitude and reliability in estimating modal frequency.

For a few decades, other researchers have studied the use of the time-difference carrier phase (TDCP) technique to determine the relative positioning vector through the GNSS system. The principal aim of this study is the application of the TDCP technique on a bridge. The main advantage of using this technique is that it does not demand high-precision products or external corrections, which is especially interesting when the Internet is not available. In addition, the TDCP technique does not depend on reference stations; therefore,

it is not affected by any local movement or deformation of those stations. Fundamentally, the TDCP technique consists of the between-epoch single difference (SD) of its carrier-phase measurements. It is vitally important to note that when the sampling interval is short, by the TDCP technique, time-correlated errors can be removed or greatly reduced. Another significant aspect of using TDCP is that it frees the process from a typical problem that has an effect on the carrier phase. If no cycle slips (CS) are produced, the ambiguity keeps constant; therefore, the ambiguity is removed by differentiating two consecutive measures.

When a temporary loss of the lock occurs in the carrier tracking loop of a GNSS receiver, it generates a CS, which is a discontinuity of the measured carrier phase. Three principal reasons engender CS: signal obstruction, an insufficient carrier/noise ratio (C/N0) parameter, and receiver software failure [22]. Julien et al. indicated receptor dynamics as a fourth reason [23]. When we perform static measurements on the ground, for quite some time, the receiver will be able to track the GPS signal continuously, so the signal changes slowly, which makes it much simpler to detect cycle slip. However, when the signal changes quickly, as in the case of the too-dynamic spatial environment, the detection of phase cycle slip becomes tricky [24].

Notably, the TDCP measurements do not have information about the absolute position and are also unprotected concerning accumulative errors.

The TDCP technique has been widely used, for example, for the velocity estimation in stand-alone GPS receivers [25]. "A few mm/s velocity accuracy is achievable based on TDCP compared to the cm/s and dm/s accuracies from Doppler measurements and differencing between positions". According to Angrisano et al. [26], the TDCP technique is extra helpful when ambiguity correction is unreliable, and specifically, this would be the case for positioning with low-cost systems. A previous study [27,28] improved positioning performance using the TDCP technique with a coupled GNSS/INS (inertial navigation system). Juan et al. [29] implemented a method based on TDCP to detect scintillation in GNSS signals. A multi-GNSS kinematic precise point-positioning (PPP) approach, which is based on the mixed-use of TDCP and undifferenced carrier-phase observations, was proposed by Yu et al. [30]. To analyse the quality of single-frequency phase data effectively and directly, Zhou et al. [31] proposed a high-rate TDCP method to study GNSS phase data quality. Ding et al. [32] determined the heading and pitch angles for vehicular applications based on TDCP; the horizontal and vertical errors were less than 1 m and 1.5 m, respectively.

CS detection and exclusion play an indispensable role in avoiding unexpected large errors. We compared two kinds of cycle-slip detector methods: the well-known Geometry-free Algorithm [22,33] and a second method, which are based on the independent term values and the residuals vector of the TDCP least squares adjustment. The aim of this study was to determine the relative positioning and associated errors of three geodetic GNSS receptors located on a bridge. According to the results, the positioning standard error was approximately 1 centimetre.

The bridge case used in this study is the Assut de l'Or bridge, which is located in Ciudad de las Artes y las Ciencias, Valencia, Spain (see Figures 1 and 2).

Bridges are crucial in the city of Valencia. Because this city is divided by a river, it remains a challenge to improve the flow between both river banks and promote growth on the left bank of the river while generating equal density between both sectors.

The Assut de l'Or bridge is a design from Santiago Calatrava that unites the original architectural design with structural singularity: a single cable-stayed span, without back spans, featuring a leaning pylon of curved geometry with stays at its top. The bridge is part of a more general architectural design, with approach spans on both sides.

This cable-stayed bridge, with only one span, has the following elements:

- Steel orthotropic deck with a span of 160 m. Its width changes between 35.5 m and 39.2 m; this last value was obtained at the crossing with the pylon.
- Steel pylon, with a height of 125 m; leaning and with a curved geometry.
- Twenty-nine main span cables in the deck.

- Four back span stays connect the top of the pylon to the concrete counterweight that forms part of one of the approach spans.

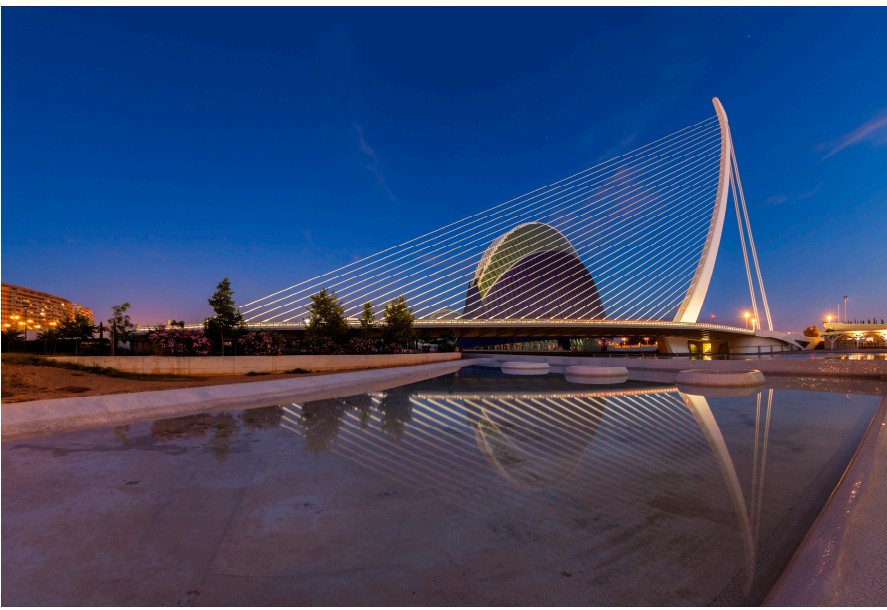

**Figure 1.** The Assut de l'Or bridge, Valencia, Spain. By Diego Delso, delso.photo, License CC-BY-SA.

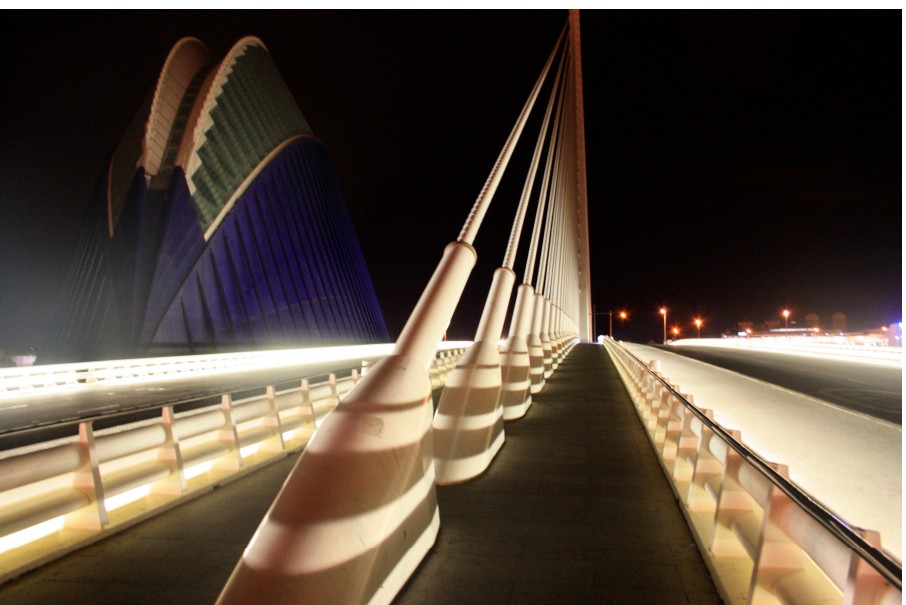

**Figure 2.** The Assut de l'Or bridge, front cable stays at night, by Frank Baulo. This image is licensed under the Creative Commons Attribution-Share Alike 3.0 Unported license.

From a functional point of view, the bridge has two carriageways with three lanes each, one additional lane for a tramway, and another laneway for pedestrian and cycle traffic located in the middle of the deck [34].

The structure of this paper will be as follows. Section 2.1 introduces the method to obtain the station displacement between adjacent epochs and associated variance from carrier-phase observations. Section 2.2 addresses the operational details to detect cycle slips, outliers, or large errors of the observations. In Section 3, the performance of the proposed method is evaluated with numerical experiments. In the final Sections 4 and 5, the discussion is drawn, and conclusions are presented.

## 2. Materials and Methods

In this section, the proposed method to obtain the relative position of a stand-alone receiver is described in detail.

Section 2.1 describes the time-differenced carrier-phase (TDCP) model, producing a continuous differential movement of the receiver. Afterward, in the second step, a detector method for cycle slips and outliers is described.

### 2.1. Time-Differenced Carrier-Phase Technique

The time-differenced carrier-phase technique denotes the difference between the two carrier-phase measurements. The satellite signal is received in the current and the previous epoch by the GNSS receptor.

Considering only two consecutive epochs for satellite $s$, the carrier phase at epoch $t_n$ is $\varphi^s_{r-tn}$, and the carrier phase in the adjacent epoch $t_{n+1}$ is $\varphi^s_{r-t(n+1)}$. Then, the two observation equations $\varphi^s_{r-tn}$ and $\varphi^s_{r-t(n+1)}$ can be written as:

$$\varphi^s_{r-tn} = \rho^s_{r-tn} + d\rho^s_{r-tn} + \lambda N^s_r + c(dt_r - dt^s) + I_{\varphi^s_r} + T_{\varphi^s_r} + dm_{\varphi^s_r} + Ep + \in_{\varphi^s_r} + r_{tn} \quad (1)$$

$$\varphi^s_{r-t(n+1)} = \rho^s_{r-t(n+1)} + d\rho^s_{r-t(n+1)} + \lambda N^s_r + c(dt_r - dt^s) + I_{\varphi^s_r} + T_{\varphi^s_r} + dm_{\varphi^s_r} + Ep + \in_{\varphi^s_r} + r_{t(n+1)} \quad (2)$$

where

$\rho^s_r$ is the distance between the receiver $r$ in reception time and the satellite $s$ in transmission time:

$$\rho^s_r = \sqrt{(X^s - X_r) + (Y^s - Y_r) + (Z^s - Z_r)}$$

$d_{\rho^s_r}$ the error in $\rho^s_r$:

$$d_{\rho^s_r} = \frac{(X^s - X_r) \cdot dx + (Y^s - Y_r) \cdot dy + (Z^s - Z_r) \cdot dz}{\rho^{s_i}_r}$$

$N^s_r$ is the integer ambiguity;
$c$ is the velocity of light;
$dt_r$ is the receiver clock error;
$dt^s$ is the satellite clock error;
$I_{PR^s_r}$ is the ionospheric delay on the GNSS signal;
$T_{PR^s_r}$ is the ionospheric delay on the GNSS signal;
$dm_{\varphi^s_r}$ is the multipath error for the carrier phase;
$Ep$ is the antenna-phase centre deviation;
$\in_{\varphi^s_r}$ is the carrier-phase measurement noise and hardware delay error;
$r_{t_n}$, $r_{t_{n+1}}$ are residual errors.

The errors of GNSS data between two adjacent epochs $t_n$ and $t_{n+1}$ tend to vary slightly, especially when observing high-rate data, such as those with a sampling rate of 0.1 s. If the rate data are high, errors have almost equal effects on the measurements and can be removed. These include atmospheric delays (as $I_{PR^s_r}$ and $T_{PR^s_r}$), satellite ephemerides errors, and clock errors.

Carrier-phase measurement noise and hardware delay errors ($\in_{\varphi^s_r}$), antenna-phase centre deviation, or multipath error for carrier phase ($Ep$) are very similar between adjacent epochs. In addition, the integer ambiguity value ($N^s_r$) will be the same if there are no cycle slips in the two adjacent epochs. The cycle slip leads to a jump in the carrier-phase measurements, usually as an integer value, but a fractional part can also occur. Section 2.2 of this paper is dedicated to the detection of cycle slips.

Aside from that, the difference of receptor–satellite distances in epochs $t_n$ and $t_{n+1}$ is $\Delta\rho^s_{t(n+1)-tn} = \rho^s_{r-t(n+1)} - \rho^s_{r-tn}$. The difference $\Delta\rho^s_{t(n+1)-tn}$ can remove most of the satellite errors and receptor coordinate errors.

Equation (1) subtracts Equation (2) to obtain:

$$\Delta\varphi^s_{t(n+1)-tn} = \varphi^s_{r-t(n+1)} - \varphi^s_{r-tn} =$$
$$\rho^s_{r-t(n+1)} - \rho^s_{r-tn} + d\rho^s_{r-t(n+1)} + cdt_{t(n+1)} - cdt_n + r_{t(n+1)} - r_{tn} = \quad (3)$$
$$\Delta\rho^s_{t(n+1)-tn} + d\rho^s_{r-t(n+1)} + \Delta cdt_{t(n+1)-tn} + \Delta r_{t(n+1)-tn}$$

where the difference of the carrier-phase residuals is $\Delta r_{2-1}$, the difference of the receiver clock offset is $\Delta cdt_{2-1}$, and $\rho^s_{r-tn} = 0$ is considered [32].

We can write Equation (3) as:

$$\Delta\varphi^s_{t(n+1)-tn} - \Delta\rho^s_{t(n+1)-tn} = d\rho^s_{r-t(n+1)} + \Delta cdt_{t(n+1)-tn} + \Delta r_{t(n+1)-tn} \quad (4)$$

The right-hand side of Equation (4) has the unknowns and the residuals.

Nonlinear equations can be linearized and solved using first-order Taylor series approximation. To obtain the linear form of Equation (4), the term $\rho^s_{r-t(n+1)}$ is derived as follows:

$$d\rho^s_{r-t(n+1)} == \left[\frac{X^s - X_r}{\rho^s_r}\right]^s_r \cdot dx^n_{n+1} + \left[\frac{Y^s - Y_r}{\rho^s_r}\right]^s_r \cdot dy^n_{n+1} + \left[\frac{Z^s - Z_r}{\rho^s_r}\right]^s_r \cdot dz^n_{n+1} \quad (5)$$

where the satellite and receiver position are $(X^s, Y^s, Z^s)$ and $(X_r, Y_r, Z_r)$. The distance between the receiver $r$ and the satellite $s$ is $\rho^s_r$.

Ding et al. [32] defined the receiver clock offset $\Delta cdt_{(n+1)-n}$ as the unknown. However, $\Delta cdt_{(n+1)-n}$ was neglected by Zhou et al. [31], so in the mathematical model, we use $\Delta cdt_{(n+1)-n} = 0$. We analyse these two proposed methods in the results section.

The left-hand side of Equation (4) has the measurements in two consecutive epochs $t_n$ and $t_{n+1}$. $\Delta\varphi^s_{(n+1)-n}$ is the difference between the two carrier-phase measurements, and $\Delta\rho^s_{(n+1)-n}$ is the difference of receptor–satellite distances.

Least squares adjustment deals with the mathematical model and the stochastic model. Equation (4) has established a relationship between parameters and observations as the mathematical model. The information regarding the precision of the measurement is introduced by the stochastic model.

The mathematical model of the initial adjustment will be the system of observation equations. The set of equations of this system is defined by Equation (4). One equation is written for each observation, satellite $s$ and two epochs.

As an example, considering two consecutive epochs $t_n$ and $t_{n+1}$ and four satellites $s1$, $s2$, $s3$, and $s4$, the system of observation equations will be:

$$\begin{aligned}
\Delta\varphi^{s1}_{t(n+1)-tn} - \Delta\rho^{s1}_{t(n+1)-tn} &= d\rho^{s1}_{r-t(n+1)} + \Delta cdt_{t(n+1)-tn} + \Delta r_{t(n+1)-tn} \\
\Delta\varphi^{s2}_{t(n+1)-tn} - \Delta\rho^{s2}_{t(n+1)-tn} &= d\rho^{s2}_{r-t(n+1)} + \Delta cdt_{t(n+1)-tn} + \Delta r_{t(n+1)-tn} \\
\Delta\varphi^{s3}_{t(n+1)-tn} - \Delta\rho^{s3}_{t(n+1)-tn} &= d\rho^{s3}_{r-t(n+1)} + \Delta cdt_{t(n+1)-tn} + \Delta r_{t(n+1)-tn} \\
\Delta\varphi^{s4}_{t(n+1)-tn} - \Delta\rho^{s4}_{t(n+1)-tn} &= d\rho^{s4}_{r-t(n+1)} + \Delta cdt_{t(n+1)-tn} + \Delta r_{t(n+1)-tn}
\end{aligned} \quad (6)$$

For performing the least squares adjustment, the above system of observation equations must be represented by the matrix notation.

We can write the system of observation equations in matrix form as [35,36]:

$$P \cdot A \cdot x = P \cdot (b + r) \quad (7)$$

Note that the system of observation equations (see example of expression (6)) is now a weighted system in Equation (7). Where the weight matrix $P$ will be diagonal because there are not correlations between consecutive epochs $t_n$ and $t_{n+1}$ [26,37,38].

The weight matrix $P$ is [26]:

$$P = \begin{bmatrix} \frac{1}{\sigma_{s1}^2} & 0 & 0 \\ 0 & \frac{1}{\sigma_{s2}^2} & 0 \\ \cdot & \cdot & \cdot \\ \cdot & \cdot & \cdot \\ \cdot & \cdot & \cdot \\ 0 & 0 & \sigma_{sn}^2 \end{bmatrix} \tag{8}$$

with the diagonal elements reflecting a weighting scheme that is a function of the satellite $s$ elevation angle [36]:

$$\sigma_s^2 = \frac{\sigma_0^2}{(satellite\ s\ elevation\ angle\ )^2} \tag{9}$$

where

$\sigma_0^2$ is a constant variance value.

Assuming that $m$ satellites are observed in two epochs $t_n$ and $t_{n+1}$, the coefficient matrix $A$ will be:

$$A = \begin{bmatrix} l_x^{s1}(t_{n+1}) & l_y^{s1}(t_{n+1}) & l_z^{s1}(t_{n+1}) & 1 \\ l_x^{s2}(t_{n+1}) & l_y^{s2}(t_{n+1}) & l_z^{s2}(t_{n+1}) & 1 \\ \cdot & \cdot & \cdot & \cdot \\ \cdot & \cdot & \cdot & \cdot \\ \cdot & \cdot & \cdot & \cdot \\ l_x^{sm}(t_{n+1}) & l_y^{sm}(t_{n+1}) & l_z^{sm}(t_{n+1}) & 1 \end{bmatrix} \tag{10}$$

where the coefficients $l_x^s$, $l_y^s$, $l_z^s$ are known from Equation (5) as:

$$l_x^s = \left[ \frac{X^s - X_r}{\rho_r^s} \right]_r^s ;\ l_y^s = \left[ \frac{Y^s - Y_r}{\rho_r^s} \right]_r^s ;\ l_z^s = \left[ \frac{Z^s - Z_r}{\rho_r^s} \right]_r^s$$

The unknowns of vector $x$ are the displacement between adjacent epochs of the receiver, and $\Delta cdt_{(n+1)-n}$ as the difference of the clock offset of the receiver:

$$x = \begin{bmatrix} dx_{n+1}^n \\ dy_{n+1}^n \\ dz_{n+1}^n \\ \Delta cdt_{(n+1)-n} \end{bmatrix} \tag{11}$$

To determine the station displacement between adjacent epochs $(dx_{n+1}^n,\ dy_{n+1}^n,\ dz_{n+1}^n)$, as a minimum, four satellites must be observed. The satellite and receiver position are also required.

The vector $b$ of the measurements is:

$$b = \begin{bmatrix} \Delta \varphi_r^{s1}(t_{n+1} - t_n) - \Delta \rho_r^{s1}(t_{n+1} - t_n) \\ \Delta \varphi_r^{s2}(t_{n+1} - t_n) - \Delta \rho_r^{s2}(t_{n+1} - t_n) \\ \cdot \\ \cdot \\ \cdot \\ \Delta \varphi_r^{sn}(t_{n+1} - t_n) - \Delta \rho_r^{sn}(t_{n+1} - t_n) \end{bmatrix} \tag{12}$$

Using matrix algebra, the least squares solution $x$ of these weighted normal equations (see Equation (7)) is:

$$x = (A^T \cdot P \cdot A)^{-1} \cdot A^T \cdot P \cdot b \tag{13}$$

The vector of residuals $r$ is:

$$r = A \cdot x - b \tag{14}$$

According to Equation (13), the vector $(dx_{n+1}^n, dy_{n+1}^n, dz_{n+1}^n)$ represents the station displacement between adjacent epochs $t_n$ and $t_{n+1}$. For high-rate GPS data, this displacement must be small. At epoch $t_{n+1}$, the vector of the station coordinates will be $x_{n+1} = x_n + dx_{n+1}^n$, $y_{n+1} = y_n + dy_{n+1}^n$, and $z_{n+1} = z_n + dz_{n+1}^n$.

The stochastic model introduces information about the precision of the adjustment. As for the precision of the results, we are using the expression of the variance–covariance matrix $\Sigma_x$. Thus, we can estimate the standard errors of the vector $x$. In the present system, the vector $x$ contains the relative coordinates of the receiver $dx_{n+1}^n$, $dy_{n+1}^n$, $dz_{n+1}^n$ and the difference of the clock offset of the receiver $\Delta cdt_{(n+1)-n}$.

As is well known, the equation of the variance–covariance matrix is:

$$\Sigma_x = \sigma_0^2 \cdot (A^T \cdot P \cdot A)^{-1} \tag{15}$$

in which the unit weight variance is:

$$\sigma_0^2 = \frac{(r^T \cdot P \cdot r)}{n - m} \tag{16}$$

where

$n$ is the number of observations;
$m$ is the number of unknowns.

The diagonal elements of the variance–covariance matrix $\Sigma_x$ are $\sigma_{xx}^2$ $\sigma_{yy}^2$, $\sigma_{zz}^2$, and $\sigma_{cdt}^2$. Where $\sigma_{xx}$, $\sigma_{yy}$, and $\sigma_{zz}$ are the standard errors of the displacements $dx_{n+1}^n$, $dy_{n+1}^n$, and $dz_{n+1}^n$, and $\sigma_{cdt}^2$ is the standard error of $\Delta cdt_{(n+1)-n}$.

These expressions provide the quality of the displacements:

$$\sigma_x = \sqrt{\sigma_{xx}^2}, \ \sigma_y = \sqrt{\sigma_{yy}^2}, \text{ and } \sigma_z = \sqrt{\sigma_{zz}^2} \tag{17}$$

For consecutive epochs, a displacement vector $(dx_t^{t-1}, dy_t^{t-1}, dz_t^{t-1})$ is obtained every two generic epochs $t_n$ and $t_{n+1}$, with the associated variance–covariance matrix $\Sigma_x$ and unit weight variance $\sigma_0^2$.

The displacement vector is the relative coordinates of the receiver $dx_{n+1}^n$, $dy_{n+1}^n$, $dz_{n+1}^n$, but these values are expressed in the geocentric coordinate system (ECEF). The last step of this subsection will be the transformation between the geocentric coordinate system (ECEF) and the east, north, and up (the vertical direction) in the local ENU coordinate system [33].

The transformation of the vector $(dx_{n+1}^n, dy_{n+1}^n, dz_{n+1}^n)$ can be described by the next equation:

$$\begin{pmatrix} \Delta E \\ \Delta N \\ \Delta U \end{pmatrix} = R(\varphi, \lambda)^T \cdot \begin{pmatrix} dx_{n+1}^n \\ dy_{n+1}^n \\ dz_{n+1}^n \end{pmatrix} \tag{18}$$

with

$$R(\varphi, \lambda) = \begin{pmatrix} -sin\lambda & -sin\varphi \cdot cos\varphi & cos\varphi \cdot cos\lambda \\ cos\varphi & -sin\varphi \cdot sin\lambda & cos\varphi \cdot sin\lambda \\ 0 & cos\varphi & sin\lambda \end{pmatrix} \tag{19}$$

where

$\varphi, \lambda$ are the latitude and longitude of the GNSS receiver, respectively.

The expressions (17) give the standard error in the geocentric coordinate system (ECEF), but, in our case, to think in terms of vertical position error ($\sigma_{\Delta Z}$) and horizontal position error ($\sigma_{\Delta E}$, $\sigma_{\Delta N}$) is more useful to understand the movement of the bridge. Thus, the transformation of the vector $\sigma_x$, $\sigma_y$, $\sigma_z$ will be:

$$\begin{pmatrix} \sigma_{\Delta E} \\ \sigma_{\Delta N} \\ \sigma_{\Delta Z} \end{pmatrix} = R(\varphi, \lambda)^T \cdot \begin{pmatrix} \sigma_x \\ \sigma_y \\ \sigma_z \end{pmatrix} \tag{20}$$

### 2.2. Cycle-Slip Detection

This subsection has described two kinds of cycle-slip detection methods that have been applied in this work. The second method is evaluated using the well-established method based on a carrier-phase geometry-free combination (GF).

#### 2.2.1. Geometry-Free Algorithm

The carrier-phase geometry-free combination (GF) needs two frequency signals. The GF combination can remove dependent effects of the frequency, such as wind-up, ionospheric delays, or instrumental delays. Additionally, measurement noise and multipath can be removed too. To estimate antenna rotations or to detect cycle slip, the GF combination is used [22,33].

The geometry-free combination can be defined as $L_{GF} = L_1 - L_2$, where $L_1$ and $L_2$ are the carrier-phase measurement at GPS frequencies of $f_1 = 1575.42$ MHz and $f_2 = 1227.6$ MHz.

The cycle slip is a jump in the carrier-phase measurements, usually as an integer value, but a fractional part can also occur. It is possible to use the geometry-free polynomial functions to detect cycle slips. These polynomial functions are determined from the last measurements. A polynomial extrapolation or polynomial interpolation can predict the carrier-phase observation. Three-point Lagrange interpolation is one of these polynomial functions. The difference between the real-phase observation and the predicted value from the polynomial function is used to detect a cycle slip. Individual cycle slip, outliers, or blunders can be detected and removed [33,39].

This type of detector has been frequently used but is not possible if some signals are momentarily unavailable.

When any of the next two conditions are satisfied, the GF detector declares a cycle slip:

- A cycle slip will be declared if the $L_{GF}$ jumps between consecutives epochs $t_t$ and $t_{t-1}$ is greater than 1 m:

$$\left| L_{GF_t} - L_{GF_{t-1}} \right| > 1 \tag{21}$$

With this condition, the proposed errors are excluded.

- This other condition consists of two statements, and both must be satisfied to detect a cycle slip:

The first statement is when the difference between $L_{GF}$ and predicted $L_{GF}$ is greater than the calculated threshold:

$$\left| L_{GF} - L_{GF-Pred} \right| > threshold_{GF} \tag{22}$$

with

$L_{GF} = L_1 - L_2$, where $L_1$ and $L_2$ are the carrier-phase measurement at GPS frequencies. $L_{GF-Pred}$ is determined with a 3-point Lagrange interpolation:

$$P(x) = \frac{(x - x_2) \cdot (x - x_3)}{(x_1 - x_2) \cdot (x_1 - x_3)} \cdot y_1 + \frac{(x - x_1) \cdot (x - x_3)}{(x_2 - x_1) \cdot (x_2 - x_3)} \cdot y_2 + \frac{(x - x_2) \cdot (x - x_1)}{(x_3 - x_2) \cdot (x_3 - x_1)} \cdot y_3 \tag{23}$$

where
$y_1$, $y_2$, and $y_3$ are the $L_{GF}$ values in the $x_1$, $x_2$, and $x_3$ epochs.
$P(x)$ is the predicted value $L_{GF-Pred}$ in epoch $x$.

The equation implemented for the $threshold_{GF}$ is:

$$threshold_{GF} = \frac{a_0}{1 + exp\left(\frac{-\Delta t}{T_0}\right)} \tag{24}$$

The maximum threshold is $a_0$, which is set as 0.08 m in our model. The minimum threshold which corresponds to $a_0/2$ is determined via $\lim_{\Delta t \to 0} 1 + exp\left(\frac{-\Delta t}{T_0}\right) = 1$. $\Delta t$ is the measurement sampling rate of 0.1 s. $T_0$ is the time decorrelation of the ionosphere, with a value of 60 s.

For the second statement, the prediction $L_{GF-Pred}$ and the observation $L_{GF}$ can be used to decide if there are any cycle slips for comparison with a residual value. $|L_{GF} - L_{GF-Pred}|$ must be greater than two times the root mean squared:

$$|L_{GF} - L_{GF-Pred}| > 2 \cdot res \tag{25}$$

The *res* value is the root mean squared of several differences between $L_{GF}$ and $L_{GF-Pred}$.

### 2.2.2. Measurement Vector $b_m$ Test for Cycle-Slip Detection

The test combines the across-epoch increments of the carrier phase and the geometric distance between the satellite and the receiver. This geometric distance between consecutive epochs is immune to cycle slips, so it can predict the variation in the carrier phase.

This procedure is based on studying the values of the vector of measurements $b_m$.

The expression of the compensated TDCP measure Equation (4) for satellite $s_m$ is:

$$\Delta \varphi^{sm}_{t-(t-1)} - \Delta \rho^{sm}_{t-(t-1)} = d\rho^{sm}_{r-(t-1)} + \Delta cdt_{t-(t-1)} + \Delta r_{t-(t-1)}$$

Given that the left-hand side of Equation (4) is the value of the measurement as the independent term $b_m$,

$$b_m = \Delta \varphi^{sm}_{t-(t-1)} - \Delta \rho^{sm}_{t-(t-1)} \tag{26}$$

If *m* satellites are observed between the two epochs $t_{tn}$ and $t_{t(n+1)}$, the compound measurement vector $b_m$ will have *m* elements for frequency $L_1$ and *m* elements for frequency $L_2$. The cycle slip does not occur simultaneously in both frequencies. This vector $b_m$ test works with each frequency, which may be an improvement.

The test declares a cycle slip if the following condition is met:

$$|b_m| > \frac{threshold_{bm}}{2} \tag{27}$$

The value of the $threshold_{bm}$ will be 0.015 m, according to deflection limits reported in Section 2 of the American Association of State Highway and Transportation Officials (AASHTO, 2020) [40]. The deflection limit is described as "the limit conditions of a structure beyond which the structure will no longer fulfil the relevant design criteria" [40]. According to guidelines published by the AASHTOT (2020), the limit value for the Assut de l'Or bridge is L/1000 when the load case involves traffic and pedestrians. In this expression, L is the length of the bridge span under consideration, and $L = 155$ m for the Assut de l'Or bridge [41].

Finally, the relative bridge movement $\Delta E$, $\Delta N$, and $\Delta U$ calculated from Equation (18) in epoch e will be excluded if:

$$\sigma_0 > p \cdot \sigma_{0-all} \tag{28}$$

where $\sigma_0$ is the standard deviation from Equation (16) in epoch e, and $\sigma_{0-all}$ is the mean of all $\sigma_0$ values from a period of n epochs with a restrictive parameter $p$, as selected by the user, for example $p = 2$.

### 3. Experiments and Results

The data description and the results, which have been obtained using the above theory, are presented in this section.

The main objective was to analyse the relative movement of three geodetic receivers placed on the Assut de l'Or bridge in València, Spain.

The Assut de l'Or bridge was officially opened to traffic in December 2008. During the current study, the bridge has been permanently open to road traffic, both cars and bicycles, and pedestrian traffic. The traffic flow data were provided by the traffic regulation department of the Valencia City Council. In this way, the traffic flow data can be temporarily correlated with the displacements measured on the bridge. This situation made it possible to evaluate the response to dynamic excitation of the bridge deck as a function of daily traffic.

In particular, the TDCP technique is used here to estimate the relative bridge movement in the north (N), east (E), and vertical (U) directions with Equations (13) and (18).

We do not include the receiver clock offset $\Delta cdt_{2-1}$ in the vector of unknowns $x$ (see Equation (4)) due to the increments of the standard deviations and the anormal graphs, especially those obtained in the vertical (U) direction. Thus, the receiver clock offset is considered null: $\Delta cdt_{2-1} = 0$.

The completed process was tested with examples.

Three double-frequency receivers Leica System 1200 (equipment manufactured by Leica and sourced in Valencia, Spain) were used for the experiment. $L_1$ and $L_2$ were the signal frequencies used. The sampling rate of GPS data was 10 Hz for a 30 min period. The data collection date was 14 December 2021. In this work, only GPS phase measurements were analysed with an elevation mask of 15°. All devices were placed on the deck in the midspan of the bridge (see Figures 3–5). Distances between receptors are shown in Table 1.

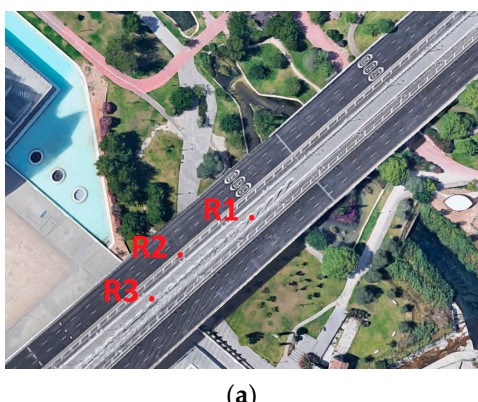

(**a**)

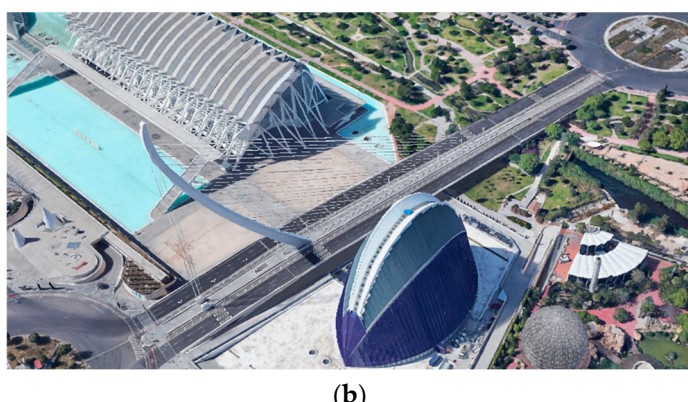

(**b**)

**Figure 3.** Positional sketch of the geodetic receivers R1, R2, and R3 (**a**), and aerial top view of the Assut de l'Or bridge (**b**).

**Table 1.** Distances between receptors R1, R2, and R3.

| Receptors | Distances (m) |
|:---:|:---:|
| R1-R2 | 17.16 |
| R2-R3 | 14.46 |
| R3-R1 | 29.03 |

To reduce the positioning error in a significant way, the sampling rate is a key factor and should be at least 10 Hz. A sampling rate of 1 Hz was not sufficient for the bridge case used in this study. The storage required for a sampling rate of 10 Hz entails substantial memory capabilities, which should also be considered.

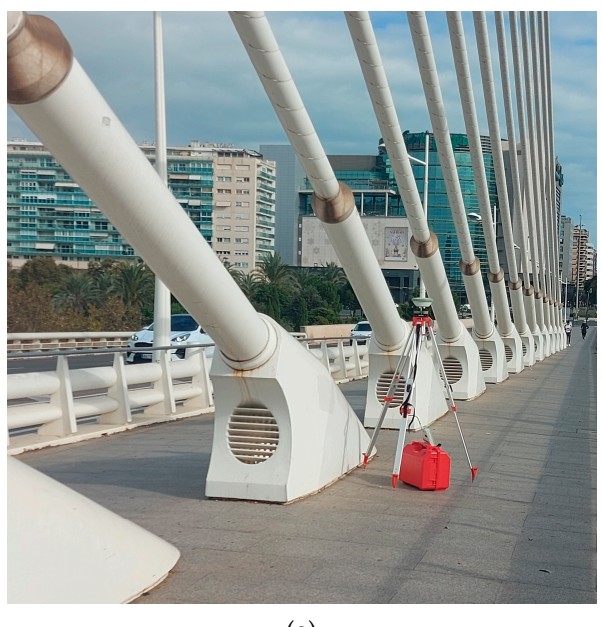
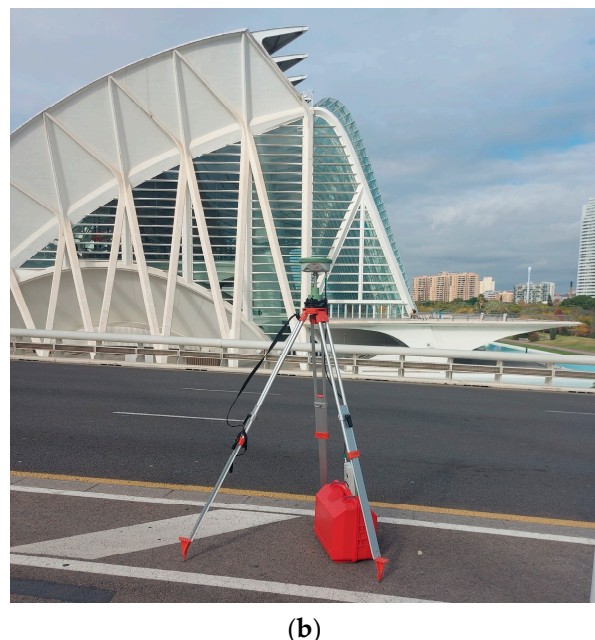

(**a**)  (**b**)

**Figure 4.** Geodetic receivers were placed on the deck in the midspan of the bridge. Geodetic receivers R1 (**a**) and R2 (**b**) were used during the survey.

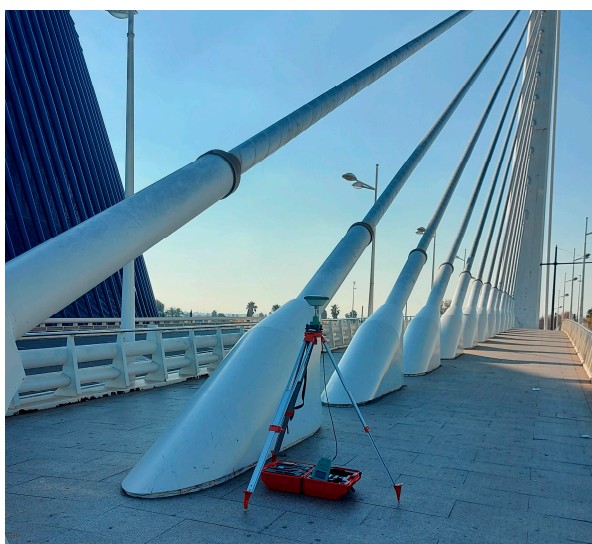

**Figure 5.** Geodetic receiver R3 during the survey.

The ephemeris data and collected observations were processed with a Phyton tool implemented by the authors. The full results were obtained by applying the developed code.

According to Equation (11), displacement vector $d_t^{t-1} = dx_t^{t-1}, dy_t^{t-1}, dz_t^{t-1}$ represents the station displacement of two generic epochs, $t_t$ and $t_{t-1}$.

$d_t^{t-1}$ must be subtracted from the previous displacement vector $d_{t-1}^{t-2}$. The difference vector $d_{t-1}^{t-2} - d_t^{t-1}$ will be the relative movement of the receiver between three epochs. The repeated subtraction processes provide all the displacements. Every subtraction can remove most satellite coordinate errors, as well as receptor coordinate errors. The generic displacement vector $d_t^{t-1}$ must be transformed into ENU coordinates according to Equation (18).

Figures 6–8 show the relative position of receptors R1, R2, and R3 in the east, north, and vertical directions.

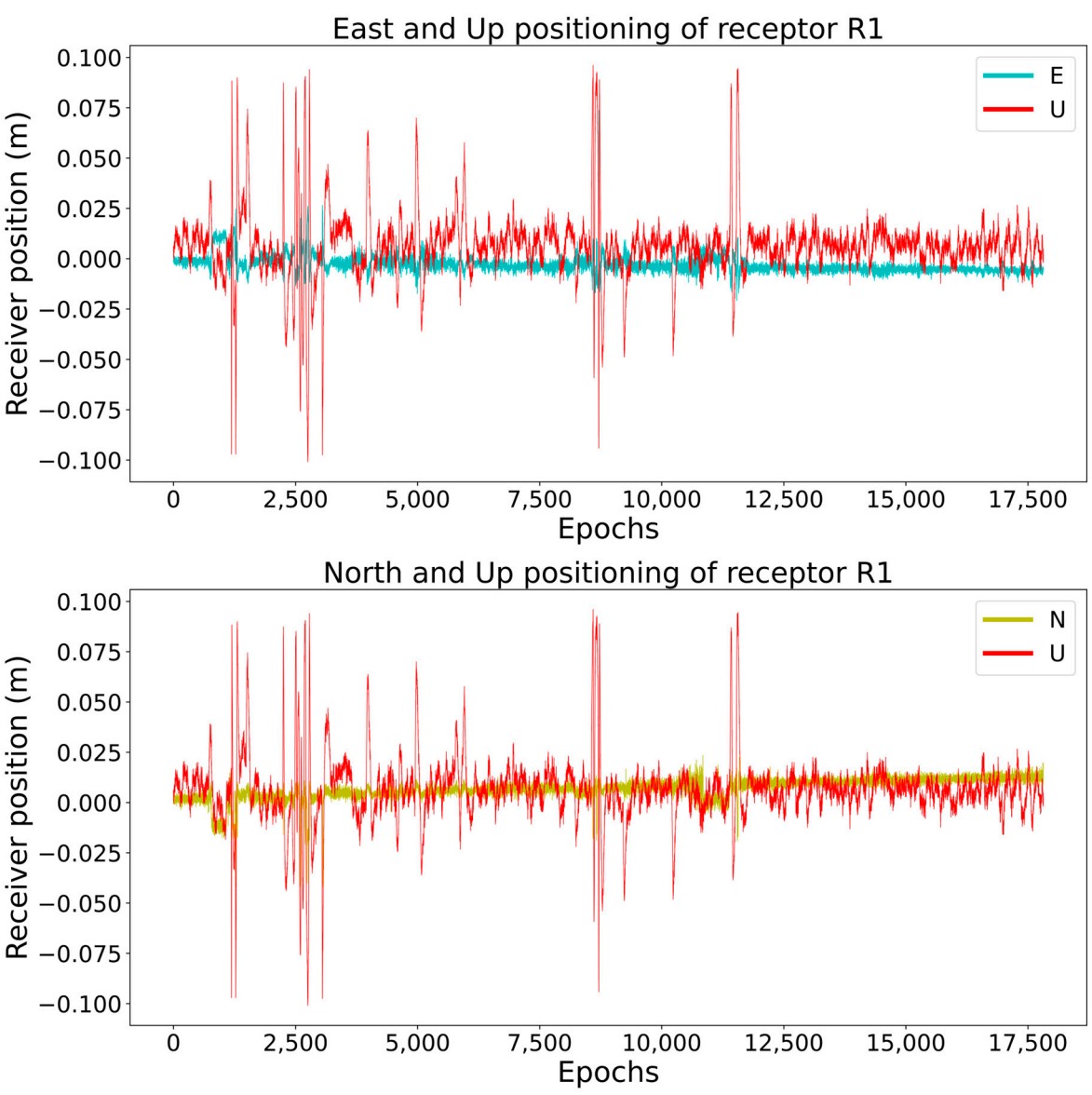

**Figure 6.** Positioning results of receptor R1 expressed in the east (E) and vertical (U) directions (**top**), as well as in the north (N) and vertical (U) directions (**bottom**). The receiver operated at 10 Hz.

The cycle-slip detector methods in Sections 2.2.1 and 2.2.2 provided similar results according to the standard deviations and movement graphs. Thus, the performance of the second method was evaluated using the well-established geometry-free (GF) method. We used the second method for all the results because this approach is better adapted to the type of processing and methodology used.

The displacements of the three receptors in the vertical (U) direction are consistent with each other, as shown in Figure 9.

Table 2 presents the mean, as the average or arithmetic mean, of the all-standard deviations associated to every position for the 30 min period. Each standard deviation is obtained from Equation (20). Means of the all-standard deviations ($\sigma_{\text{E−Mean}}\sigma_{\text{E−Mean}}$, north $\sigma_{\text{N−Mean}}$, and up $\sigma_{\text{U−Mean}}$) associated to every position for a 30 min period.

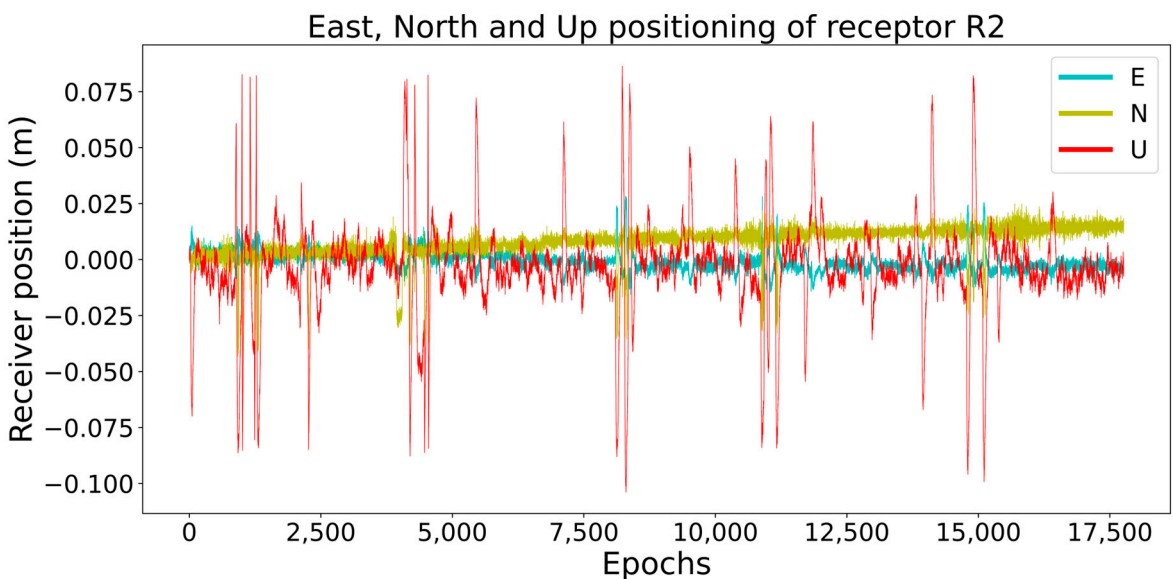

**Figure 7.** Positioning results of receptor R2 in ENU coordinates. The east €, north (N), and vertical (U) directions. The receiver operated at 10 Hz.

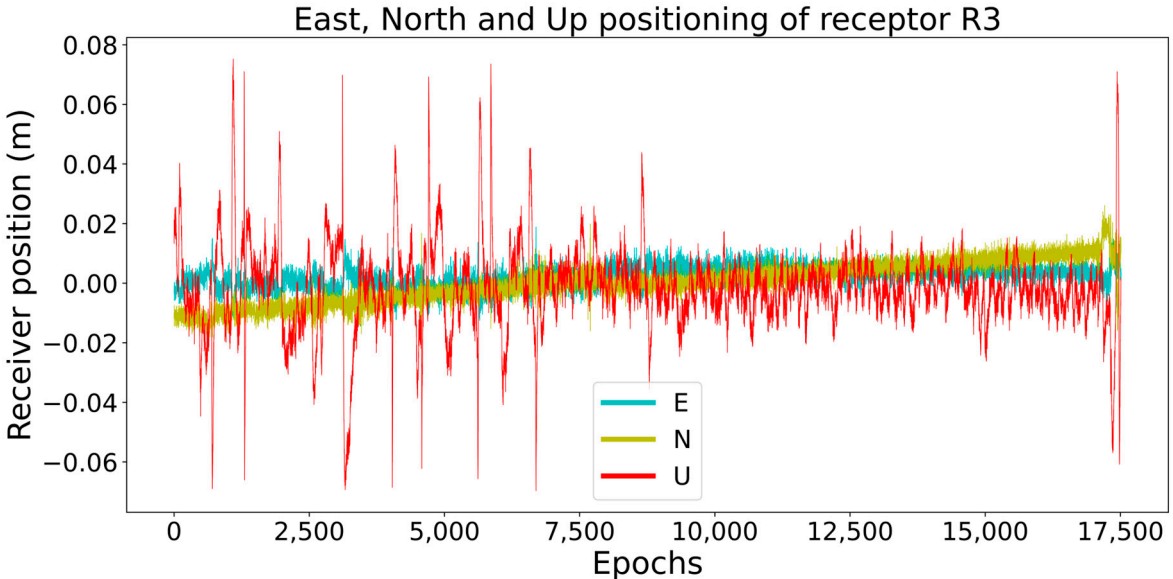

**Figure 8.** Positioning results of receptor R3 in ENU coordinates. The east (E), north (N), and vertical (U) directions. The receiver operated at 10 Hz.

**Table 2.** The maximum displacements of the TDCP solution ($\Delta$E-TDCP, $\Delta$N-TDCP, $\Delta$U-TDCP), and means of the all-standard deviations associated to every displacement ($\sigma_{E-Mean}\sigma_{E-Mean}$, north $\sigma_{N-Mean}$, and up $\sigma_{U-Mean}$), in a 30 min period. Standard deviations are obtained from Equation (20).

| Receptor | Epochs | $\Delta$E-TDCP (m) | $\Delta$N-TDCP (m) | $\Delta$U-TDCP (m) | $\sigma_{E-Mean}$ (m) | $\sigma_{N-Mean}$ (m) | $\sigma_{U-Mean}$ (m) |
|---|---|---|---|---|---|---|---|
| **R1** | 17,775 | 0.043 | 0.059 | 0.196 | 0.0023 | 0.0020 | 0.0068 |
| **R2** | 17,770 | 0.043 | 0.068 | 0.190 | 0.0019 | 0.0021 | 0.0072 |
| **R3** | 17,832 | 0.061 | 0.049 | 0.149 | 0.0012 | 0.0012 | 0.0040 |

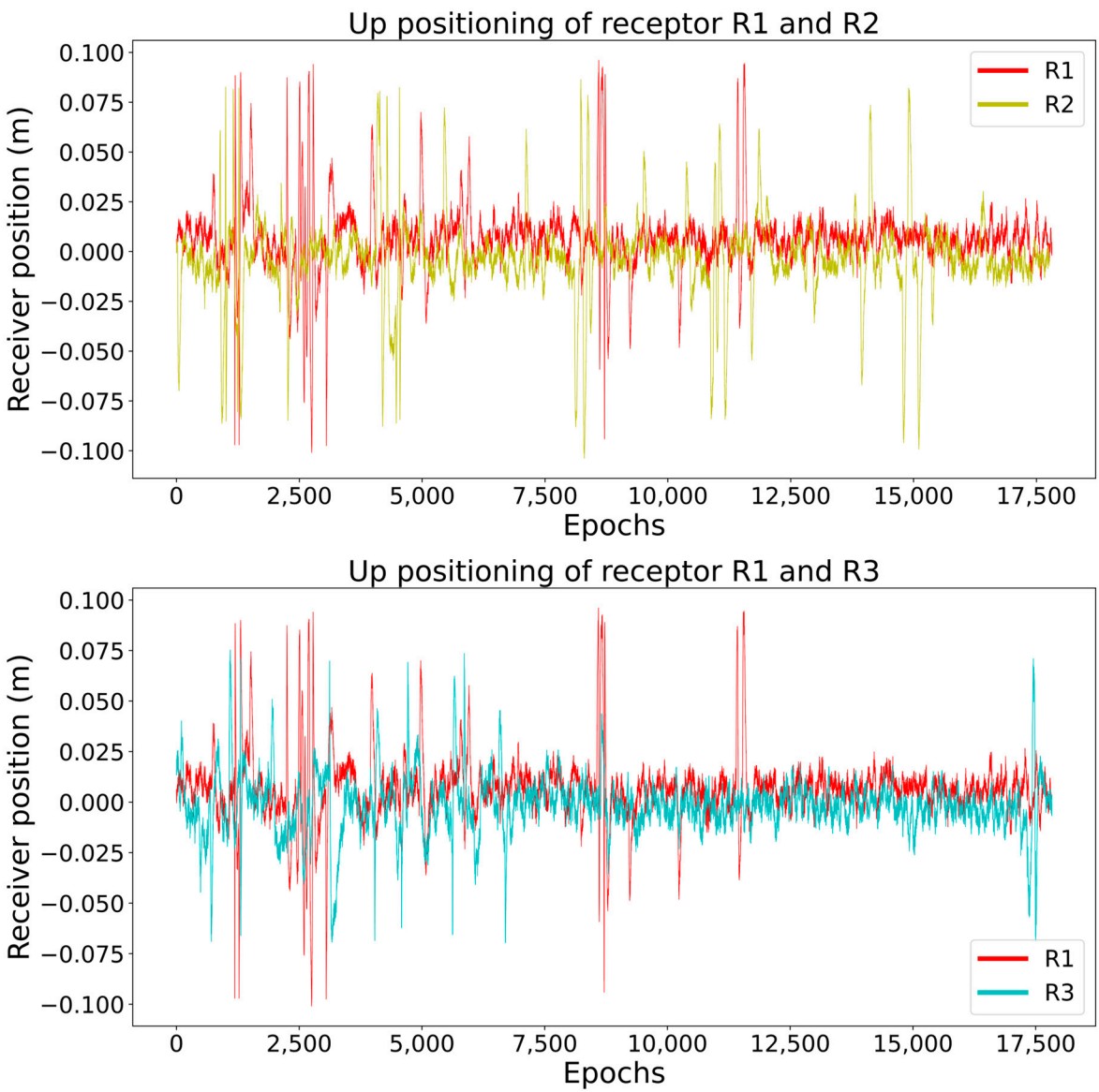

**Figure 9.** Positioning in the vertical (U) direction of receptors R1 and R2 (**top**) and receptors R1 and R3 (**bottom**). The receiver operated at 10 Hz.

Real-Time Kinematic (RTK) is a positioning technique that uses carrier-phase and pseudo-range measurements. The position calculations are performed using the base station and the rover. In this study, R1, R2, and R3 receivers are the rovers. The data observation with this methodology requires the base station, which is working simultaneously with the rover receivers and is located outside the bridge. RTK is used for applications that require higher accuracies, such as centimetre-level positioning, up to 1 cm + 1 ppm.

The professional software Leica Infinity was used to obtain a post-processed RTK solution of the relative displacements of the bridge and associated standard deviation in the ENU local system.

The RTK method is less accurate than the TDCP, but we could not find a better kinematic solution to compare to the TDCP solution. Tables 2 and 3 show the means, as averages or arithmetic means, of the all-standard deviations associated with every position for a 30 min period. The TDCP solution has a standard deviation means of approximately 0.005 m, and the RTK solution has a standard deviation means of approximately 0.015 m.

**Table 3.** The maximum displacements of the post-processed RTK solution (ΔE-RTK, ΔN-RTK, ΔU-RTK), and the means of the all-standard deviations associated with every movement ($\sigma_{E-Mean}$, $\sigma_{N-Mean}$, and $\sigma_{U-Mean}$), for a 30 min period. Displacements and associated standard deviations were obtained using the professional software Leica Infinity.

| Receptor | ΔE-RTK (m) | ΔN-RTK (m) | ΔU-RTK (m) | $\sigma_{E-Mean}$ (m) | $\sigma_{N-Mean}$ (m) | $\sigma_{U-Mean}$ (m) |
|---|---|---|---|---|---|---|
| R1 | 0.054 | 0.059 | 0.198 | 0.019 | 0.015 | 0.024 |
| R2 | 0.022 | 0.060 | 0.149 | 0.021 | 0.008 | 0.022 |
| R3 | 0.041 | 0.043 | 0.180 | 0.018 | 0.009 | 0.019 |

The maximum displacements obtained from the TDCP technique were evaluated against the maximum displacements of the post-processed RTK solution. The maximum displacements obtained from the TDCP technique were evaluated against the maximum displacements of the post-processed RTK, as a well-known method. The results of both methods are very similar. The largest displacement of the bridge is the vertical direction (see ΔU-TDCP values in Table 2 and ΔU-RTK values in Table 3). The ΔU-TDCP values for R1, R2, and R3 receptors were 0.196 m, 0.190 m, and 0.0149 m, respectively. The ΔU-RTK values for R1, R2, and R3 receptors were 0.198 m, 0.149 m, and 0.018 m, respectively. Thus, the TDCP solution is in accordance with the RTK solution.

Recovering the initial position after movement is a key factor in the maintenance and detection of the potential structural problems of a bridge. To detect that recovering, we have obtained the means of all displacements in a 30 min period by using the TDCP solution. If the means, as averages, are close to 0 m, the recoverable deformation in the period time could be evidenced. Table 4 presents the means of the east (ΔE − Mean ), north (ΔN − Mean ), and vertical (ΔU − Mean ) displacements, all of which are lower than 0.004 m. It could be said that for most of the time, the bridge remains in the zero position.

**Table 4.** Means, as averages or arithmetic means, of all displacements in a 30 min period: east (ΔE − Mean ), north (ΔN − Mean ), and vertical (ΔU − Mean ) directions.

| Receptor | ΔE − Mean (m) | ΔN − Mean (m) | ΔU − Mean (m) |
|---|---|---|---|
| R1 | 0.004 | 0.004 | 0.003 |
| R2 | 0.001 | 0.002 | −0.003 |
| R3 | 0.002 | 0.001 | −0.001 |

## 4. Discussion

The main objective of this study was to analyse relative bridge deformation using a stand-alone GNSS receiver; in particular, the TDCP method was considered. An advantage of working with this technique is its lack of a need for external corrections or high-precision products, especially when the Internet is not available. The TDCP technique is also independent from the reference stations; therefore, it is not affected by any local movement or deformation of those stations, and consequently, it needs less GNSS equipment.

For comparison, a differential kinematic GNSS solution was obtained to assess the positioning results. The standard deviation of the RTK solution was found to be low (see Table 3), which was expected under this methodology, but we could not find a better kinematic solution to estimate the quality of the TDCP solution in this article.

According to the results of the proposed TDCP method, the positioning movement was estimated to be precise. The means of the standard deviation of the east, north, and vertical directions were approximately 0.003, 0.003, and 0.008 m (see Table 2), for a 30 min period. This method has a precision equal to or inferior to other methods currently used, such as RTK.

The exclusion of cycle slips in the TDCP method was essential to improve the accuracy. The performance of the vector $b_m$ test (see Section 2.2.2) was evaluated using the well-

established detector based on the carrier-phase geometry-free combination (GF). As both methods offered very similar results in this work, we used the second one. The value of *threshold*$_{bm}$ according to deflection limits reported in AASHTO (2020) was key for the success of this method.

The vector $b_m$ test can detect the carrier frequency with cycle slip. This detector works independently for each signal frequency, which may be an improvement. The cycle slip does not occur simultaneously in all frequencies. L1 frequency may be less affected by cycle slips than the L2 frequency [42].

Cycle slips must be removed, so a cycle-slip repair technique would improve the TDCP performance. The TDCP mode cannot work correctly if the solutions are not continuous. Only GPS carrier phases were used in this study, as adding other satellite systems could avoid discontinuities in signal reception. Triple-frequency GNSS receptors could help improve cycle-slip detection and improve data quantity [43], such receptors could be studied in future research.

Research revealed that the response of the bridge is affected by the characteristics of the bridge as the natural frequencies of the bridge or mass distribution. Behind that, the characteristics of the vehicles and t bridge surface can affect the dynamic response of the bridge. These different parameters interact with each other and render it more difficult to study the dynamic response of a bridge, as reported by the National Cooperative Highway Research Program (NCHRP). Nevertheless, in the existing American Association of State Highway and Transportation Officials (AASHTO) Standard Specifications for Highway Bridges, the dynamic load is a function of bridge span only [7].

A suspension bridge has different kinds of deformations. One of these is slow, long-term, and unrecovered movement. The second type is short-term displacement or movement, which is produced by factors such as traffic or wind [41]. It is precisely this short-term and recoverable movement that is evidenced by the displacement graphs obtained in this approach, and the importance of recovering the initial position after a deformation is remarkable. Knowing the behaviour of a bridge from displacement graphs from a long period of time might reveal an accurate response of the bridge, making it easier to detect structural problems. "The risk of bridge failures cannot be eliminated, a good maintenance program including regular inspection will slow down the deterioration process of bridges and help detect potential structural problems before they develop into serious disasters" [44].

As previously noted, a higher data sampling frequency is a key factor for improving accuracy. The errors of GNSS data between two adjacent epochs $t_n$ and $t_{n+1}$ tend to vary slightly, especially when observed at a 10 Hz sampling rate. If the rate data are high, errors have almost equal effects on the two measurements and can be removed. These include atmospheric delays, satellite coordinate errors, and clock errors. In addition, carrier-phase measurement noise and hardware delay error, antenna-phase centre deviation, or multipath error are very similar between adjacent epochs. In addition, the integer ambiguity value will be the same if cycle slips are excluded. Future researchers should study the sampling rate of 100 Hz, chiefly in kinematic mode.

In a static survey, the GNSS receiver tracks satellite signals continuously, so the carrier phase changes slowly, making the detection of cycle slips easier. However, in the dynamic survey, the carrier phase changes rapidly, making it more difficult to detect cycle slips [24]. It must be taken into account if the TDCP methodology is applied in a highly dynamic environment.

The TDCP model used in this study to control bridge movement could be expanded to single and double differences in carrier-phase measurements, such as relative positioning with two or more stations [45]. Single and double differences could facilitate cycle-slip detection, improve accuracy, and reinforce the confidence in the results. Additionally, the least squares network could be assisted by external geometric conditions, which could improve the compensated network solution and characterize the project [46,47].

The study of positioning techniques based on sensor fusion could provide actionable information in the present and near future. The TCDP high-rate method could be

used with inertial or vision sensors detecting the movements and starting the control of the displacements.

It is important to highlight the generality of the proposed methodology, which could be applied to other civil structures.

A limitation or disadvantage of the TDCP algorithm is the requirement of a high sampling rate of the receivers. Operating at 10 Hz is essential for the success of the method. In this way, the errors in such close epochs are very similar and, by subtracting carrier phases, can be eliminated. Geodetic receivers can record data at 10 Hz or even more in some cases.

A sampling rate of 1 Hz was not enough for the bridge case because the standard error was approximately 0.09 m (obtained from expression (17)), and the size of the displacements was very similar.

The sampling rate is correlated with the level of accuracy needed in the project. In our case, the level of accuracy selected was 0.01 m. Expression (17) makes it possible to know the standard deviations of the TDCP method. The standard deviations will be compared to the level of accuracy needed in each project.

Presently, it is not possible to use mass-market GNSS devices with the TDCP technique because their sampling rate is lower than 10 Hz. However, the current trend is to use low-cost GNSS sensors, especially with the Internet of Things (IoT). IoT allows permanent control of the structures 7 days per week and 24 h per day to detect anomalies and to have a precise model of the bridge movements. Thus, a superior quantity of data will improve the accuracy of the mean measurements.

Another limitation of the TDCP algorithm is the necessary exclusion of cycle slips. Each project will have to find the best-adapted algorithms to detect them. As an example, in this study, the deflection value of the bridge provided a threshold to declare a cycle slip. Many methods are known, but not all are effective for the structure of the project. If no effective method is found, the TDCP algorithm will not work.

## 5. Conclusions

The final goal of the TDCP algorithm was to determine the relative positioning and associated standard deviation of a stand-alone GNSS geodetic receiver in the local ENU coordinate system. The mean of the standard deviations for a 30 min period of the east, north, and vertical relative positions (ENU) was approximately 0.003 m, 0.003 m, and 0.008 m, respectively. Results were evaluated against a post-processed differential kinematic GNSS solution.

The TDCP algorithm together with the method for cycle-slip detection was tested in three stand-alone receivers located on the Assut de l'Or bridge, València, Spain. Three geodetic receivers of the Leica System 1200 device were used to collect data.

The use of a 10 Hz sampling rate was a key factor in reducing the positioning error in a significant way.

The deflection value of the bridge provided a threshold to declare a cycle slip.

The continuous control of movement could help detect structural problems, preventing bridges from turning into a tragedy or undergoing progressive deterioration. The short-term movement produced by traffic, wind, or earthquakes must be a recoverable deformation. The short-term movement of the structure is evidenced by the displacement graphs obtained in this paper. The mean of the ENU displacements confirms that there is a return to the starting positioning. Thus, the recoverable deformation is evidenced by the TDCP method.

**Author Contributions:** Conceptualization, M.J.J.-M.; methodology, M.J.J.-M.; software, M.J.J.-M.; validation, M.J.J.-M. and N.Q.-O.;formal analysis, M.J.J.-M.; investigation, M.J.J.-M., N.Q.-O., J.J.Z.-J. and T.M.-P.; resources, M.J.J.-M., N.Q.-O., J.J.Z.-J. and T.M.-P.; data curation, M.J.J.-M.; writing—original draft preparation, M.J.J.-M. and N.Q.-O.; writing—review and editing, M.J.J.-M., N.Q.-O., J.J.Z.-J. and T.M.-P.; visualization, M.J.J.-M., N.Q.-O., J.J.Z.-J. and T.M.-P.; supervision, J.J.Z.-J. and T.M.-P.; project administration, N.Q.-O.;funding acquisition, N.Q.-O. All authors have read and agreed to the published version of the manuscript.

**Funding:** This research was funded by Generalitat Valenciana, grant number GV/2021/156.

**Data Availability Statement:** Not applicable.

**Acknowledgments:** The authors are grateful to the Editor and anonymous reviewers for their valuable suggestions and constructive comments. They have provided vital insights and helped the authors shape the early drafts of the paper into their final form.

**Conflicts of Interest:** The authors declare no conflict of interest. The founding sponsors had no role in the design of the study; in the collection, analyses, or interpretation of data; in the writing of the manuscript, and in the decision to publish the results.

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
