# Peer review of "Bridge Deformation Analysis Using Time-Differenced Carrier-Phase Technique"

_remotesensing, doi:10.3390/rs15051458_

Round 1

Reviewer 1 Report

In line 592 is an error in the referencing

In Chapter 3 explain if there was any loading of the bridge by traffic or if the bridge was unloaded.
Were there tests with a loading vehicle? What are the results and how plausible is this?

Author Response

Response to Reviewer 1 Comments

First of all, we would like to thank the Editor and the Reviewers for their careful reading of the manuscript and the significant effort they have made. Reviewers have provided an in-depth review and given us many tips to improve the formal aspects, wording, and terminology of the manuscript. We would especially like to thank reviewer 1; thanks to their valuable view we have been able to clarify concepts and correct inaccuracies.

We have included all the suggestions proposed by the reviewer. Please, find replies to every individual comment below (in red).

We also provide a marked version of the corrected manuscript to quickly detect the new contents. Note that blue text refers to changes.

We have corrected the wording and translation for a better understanding.

Point 1: In line 592 is an error in the referencing

Response 1: We have adopted the suggestion of the reviewer and the referencing error has already been corrected.

Point 2: In Chapter 3 explain if there was any loading of the bridge by traffic or if the bridge was unloaded.
Were there tests with a loading vehicle? What are the results and how plausible is this?

Response 2: We have adopted the suggestion of the reviewer and we have added these lines in chapter 3 (in blue):

The Assut de l'Or bridge is officially open to traffic in December 2008. During the current study, the bridge is permanently open to road traffic, car and bicycle, and pedestrian traffic. The traffic flow data was provided by the traffic regulation section of the Valencia City Council. In this way, the traffic flow data can be temporarily correlated with the displacements measured on the bridge. This situation made it possible to evaluate the response to dynamic excitation of the bridge deck as a function of daily traffic.

Were there tests with a loading vehicle? What are the results and how plausible is this?

The reception load tests of a structure are mandatory according to the "Instruction on the actions to be considered in the road bridge project (IAP-11)", approved in September 2011 by the Ministry of Public Works of Spain. Its objective is to confirm that the design and construction of the project have been executed satisfactorily and must be carried out before the bridge will be open to traffic.

The "Recommendations for carrying out reception load tests on highway bridges" of Directorate General of Highways. Ministry of Public Works, in chapter 4: Approach of the load test, determine that the approach of the reception load test must be included in the Project of the structure and drawn up by the author of the same. In this same chapter, the measurement systems that must be used for the evaluation of the deflections produced by the applied static loads are specified so that they have a different evaluation methodology from the one proposed in this paper.

The displacements of the three receptors in the vertical (U) direction are according to deflection limits reported in Section 2 of the American Association of State Highway and Transportation Officials (AASHTO, 2020) and in section 7. Criteria for checking limit states of deformations the "Instruction on the actions to be considered in the road bridge project” (IAP-11).

Reviewer 2 Report

The manuscript has a relatively acceptable flow and structure overall. It also seems to have an interesting topic for the readers; however, there are a number of issues throughout the entire manuscript that need to be fixed. The following points are some of the most significant issues that this reviewer recommend the authors to take into account to enhance the quality and writing of the manuscript:

The authors may need to use and refer to some of the most recent publications on the topic of this manuscript. There are only a few recently published studies among the existing references. The authors should further emphasize on the novelty of their research by providing additional recent references and making pair-wise comparison to prove their contribution to the body of knowledge.

The mathematical part in Section 2, i.e., materials and methods, need further elaborations; most of the provided formulas require additional explanation. The current way the authors developed this section is ambiguous to some degree, especially for the readers who are not fully familiar with structural concepts.

The authors may need to elaborate the conceptual justifications behind the formulas provided in Section 2, i.e., materials and methods, to make it more sensible for the readers.  

The quality of Figures 6 through 9 is not sufficient for the purpose of publication in this journal. The authors may need to provide the high-quality version of these figures.

There are a couple of unintended writings throughout the manuscript which should be removed, such as in Lines 576 and 592.

The authors may need to further mention the limitations of their study in the discussion section of the manuscript.

Author Response

Response to Reviewer 2 Comments

First of all, we would like to thank the Editor and the Reviewers for their careful reading of the manuscript and the significant effort they have made. Reviewers have provided an in-depth review and given us many tips to improve the formal aspects, wording, and terminology of the manuscript. We would especially like to thank reviewer 2; thanks to their valuable view we have been able to clarify concepts and correct inaccuracies.

We have included all the suggestions proposed by the reviewer. Please, find replies to every individual comment below (in red).

We also provide a marked version of the corrected manuscript to quickly detect the new contents. Note that blue text refers to changes.

We have corrected the wording and translation for a better understanding.    

Point 1:

The manuscript has a relatively acceptable flow and structure overall. It also seems to have an interesting topic for the readers; however, there are a number of issues throughout the entire manuscript that need to be fixed. The following points are some of the most significant issues that this reviewer recommends the authors to take into account to enhance the quality and writing of the manuscript.

The authors may need to use and refer to some of the most recent publications on the topic of this manuscript. There are only a few recently published studies among the existing references. The authors should further emphasize the novelty of their research by providing additional recent references and making pairwise comparison to prove their contribution to the body of knowledge.

Response 1: We have adopted the suggestion of the reviewer and we refer to some of the most recent publications on the topic of this manuscript. We have modified Section 1. Introduction, in the marked version of the corrected manuscript, can be detected with the new contents in blue.

Point 2:

The mathematical part in Section 2, i.e., materials and methods, need further elaborations; most of the provided formulas require additional explanation. The current way the authors developed this section is ambiguous to some degree, especially for the readers who are not fully familiar with structural concepts.

The authors may need to elaborate the conceptual justifications behind the formulas provided in Section 2, i.e., materials and methods, to make it more sensible for the readers. 

Response 2: We have modified Section 2 (Materials and Methods), for a better understanding and greater clarity of the mathematical and statistical methods, adding detailed additional explanations. In the marked version of the corrected manuscript can be detected the new contents in blue.

Point 3:

The quality of Figures 6 through 9 is not sufficient for the purpose of publication in this journal. The authors may need to provide the high-quality version of these figures.

Response 3: Figures 6 to 9 have been improved. All of them with a dimension of 1200 dpi, which provides a high resolution.

Point 4:

There are a couple of unintended writings throughout the manuscript which should be removed, such as in Lines 576 and 592.

Response 4: We have removed the unintended references.

Point 5:

The authors may need to further mention the limitations of their study in the discussion section of the manuscript.

Response 5: It has been added these lines at the end of the discussion section:

A limitation or disadvantage of the TDCP algorithm is the requirement of a high sampling rate of the receivers. Operating at 10 Hz is essential for the success of the method. In this way, the errors in such close epochs are very similar and, by subtracting carrier phases, are eliminated. Geodetic receivers can record data at 10 Hz or even more in some cases.

A sampling rate of 1 Hz was not enough for the bridge because the standard error was around 0.09 m (from expression (17)), and the size of the displacements was similar.

The sampling rate is correlated with the level of accuracy needed in the project. In our case, the level of accuracy was 0.01 m. Expression (17) allows knowing the standard deviations of the TDCP method. The standard deviations will be compared to the level of accuracy needed in each project.

Nowadays it is not possible to use mass-market GNSS devices with TDCP technique because their sampling rate is lower than 10 Hz. However, the current trend is to use low-cost GNSS sensors, especially with the Internet of Things (IoT). IoT allows permanent control of the structures, 7 days 24 hours, to detect anomalies and to have a precise model of the bridge movements. Being that a superior quantity of data will improve the accuracy of the mean measurements.

Another limitation of the TDCP algorithm is the necessary exclusion of cycle slips. Each project will have to find the best-adapted algorithms to detect them. As an example, in this study, the deflection value of the bridge provided a threshold to declare a cycle slip. Many methods are known, but not all are effective for the structure of the project. If no effective method is found the TDCP algorithm will not work.

Reviewer 3 Report

The paper deals with studying of the movement of a long-span bridge structure by using different carrier phases in adjacent epochs. One of the advantages is that two receivers observing simultaneously to determine relative displacements are not required. The final goal of this paper is to obtain relative positioning and associated standard deviations of a stand-alone geodetic receiver. The paper describes an innovative approach for analysis of deformation. Thus, the paper can be published.  

Author Response

Response to Reviewer 3 Comments

The paper deals with studying of the movement of a long-span bridge structure by using different carrier phases in adjacent epochs. One of the advantages is that two receivers observing simultaneously to determine relative displacements are not required. The final goal of this paper is to obtain relative positioning and associated standard deviations of a stand-alone geodetic receiver. The paper describes an innovative approach for analysis of deformation. Thus, the paper can be published.  

Submission Date

16 January 2023

Date of this review

29 Jan 2023 17:38:15

 Response 1: First of all, we would like to thank the Editor and the Reviewers for their careful reading of the manuscript and the significant effort they have made.

We would especially like to thank reviewer 1; thanks to having provided an in-depth review and have trusted our work.

Even so, we have corrected the wording and translation for a better understanding.

Reviewer 4 Report

The paper presented a research on the deformation measurement method for bridge structures, which is interesting subject for the structural health monitoring field, in which the structural deformation is an important index for evaluate the structural performance. Before further consideration about publication, the follow concerns should be responded and the revised one should be returned to the reviewer for re-review.

1.       In the manuscript, the authors propose a method for analyzing bridge deformation using time- differenced carrier phases technique. However, for the structural health monitoring (SHM) of a practical bridge, what are the main advantages and technological improvements of the proposed method (focus on contributions to bridge monitoring)? And Compared with the existing widely used methods, would it have a significant improvement in measurement accuracy for he proposed method? Relevant descriptions and comparisons are not obvious in this study. Please explain it and revise the manuscript if considered necessary.

2.       Since the manuscript presented a subject about bridge deformation measurement which is closely related to structural health monitoring, the literature and comments about this field should be added to the introduction part, such as

“Correlation-based estimation method for cable-stayed bridge girder deflection variability under thermal action, Journal of Performance of Constructed Facilities, 2018, 32(5)”

“Monitoring and analysis of thermal effect on tower displacement in cable-stayed bridge, Measurement, 2018, 115(2018)”

3.       Lines 431-432: Would it have a basis for the sampling rate of 10 Hz? Whether the value of sampling rate should be changed for different practical bridges? If so, could you describe a general method to determine the value of the sampling rate, or it necessarily needs to be selected subjectively? Please explain it.

4.       Line 512: The “RTK” does not appear in the previous sections. Please explain its meaning here and revise the manuscript.

5.       Lines 576 and 592: Please check and correct the errors in the manuscript.

6.       Figures 6-9: The conclusion that " Recover the initial position after a movement " is not clear and obvious in Figure 6-9. Please describe it in detail and revise the manuscript if considered necessary.

7.       Tables 2 and 4: What is the basis for evaluating the RTK and TDCP methods based on the standard deviation?

8.       Vehicle-induced displacement response is vital for bridge monitoring. Could you describe whether the proposed method can accurately capture the vehicle excitation and measure the vehicle-induced displacement? If so, what level of measurement accuracy can be achieved?

Author Response

Response to Reviewer 4 Comments

First of all, we would like to thank the Editor and the Reviewers for their careful reading of the manuscript and the significant effort they have made. Reviewers have provided an in-depth review and given us many tips to improve the formal aspects, wording, and terminology of the manuscript. We would especially like to thank reviewer 4; thanks to their valuable view we have been able to clarify concepts and correct inaccuracies.

We have included all the suggestions proposed by the reviewer. Please, find replies to every individual comment below (in red).

We also provide a marked version of the corrected manuscript to quickly detect the new contents. Note that blue text refers to changes.

We have corrected the wording and translation for a better understanding. 

Point 1:

In the manuscript, the authors propose a method for analyzing bridge deformation using time- differenced carrier phases technique. However, for the structural health monitoring (SHM) of a practical bridge, what are the main advantages and technological improvements of the proposed method (focus on contributions to bridge monitoring)? And Compared with the existing widely used methods, would it have a significant improvement in measurement accuracy for the proposed method? Relevant descriptions and comparisons are not obvious in this study. Please explain it and revise the manuscript if considered necessary.

Response 1:

In the discussion section, it can be read these possible improvements, we have added some words (here in red bold):

1- An advantage of working with this technique is that its lack of a need for external corrections or high-precision products, especially when the Internet is not available. The TDCP technique is also independent from the reference stations; therefore, it is not affected by any local movement or defor-mation of those stations, and consequently, it needs less GNSS equipment.

2- According to the results of the proposed TDCP method, the positioning movement was estimated to be precise. The means of the standard deviation of the east, north, and vertical directions were around 0.003, 0.003, and 0.008 m.

Afterward the text before we added these new lines to the manuscript:

This method has a precision equal to or inferior to other methods currently used, such as RTK.

We have added some limits or disadvantages of the TDCP algorithm at the end of the discussion section:

A limitation or disadvantage of the TDCP algorithm is the requirement of a high sampling rate of the receivers. Operating at 10 Hz is essential for the success of the method. In this way, the errors in such close epochs are very similar and, by subtracting carrier phases, are eliminated. Geodetic receivers can record data at 10 Hz or even more in some cases.

A sampling rate of 1 Hz was not enough for the bridge because the standard error was around 0.09 m (from expression (17)), and the size of the displacements was similar.

The sampling rate is correlated with the level of accuracy needed in the project. In our case, the level of accuracy was 0.01 m. Expression (17) allows knowing the standard deviations of the TDCP method. The standard deviations will be compared to the level of accuracy needed in each project.

Nowadays it is not possible to use mass-market GNSS devices with TDCP technique because their sampling rate is lower than 10 Hz. However, the current trend is to use low-cost GNSS sensors, especially with the Internet of Things (IoT). IoT allows permanent control of the structures, 7 days 24 hours, to detect anomalies and to have a precise model of the bridge movements. Being that a superior quantity of data will improve the accuracy of the mean measurements.

Another limitation of the TDCP algorithm is the necessary exclusion of cycle slips. Each project will have to find the best-adapted algorithms to detect them. As an example, in this study, the deflection value of the bridge provided a threshold to declare a cycle slip. Many methods are known, but not all are effective for the structure of the project. If no effective method is found the TDCP algorithm will not work.

Point 2:

Since the manuscript presented a subject about bridge deformation measurement which is closely related to structural health monitoring, the literature and comments about this field should be added to the introduction part, such as

“Correlation-based estimation method for cable-stayed bridge girder deflection variability under thermal action, Journal of Performance of Constructed Facilities, 2018, 32(5)”

“Monitoring and analysis of thermal effect on tower displacement in cable-stayed bridge, Measurement, 2018, 115(2018)”

Response 2: We have adopted the suggestion of the reviewer and we have added new references in Chapter 1. Introduction:

Within the same field of structural health monitoring, interesting studies related to the influence of thermal action and displacement in cable-stayed bridges are being developed in parallel:

-Yang, Dong-Hui & Yi, Ting-Hua & Li, Hong-Nan & Zhang, Yu-Feng. (2018). Correlation-Based Estimation Method for Cable-Stayed Bridge Girder Deflection Variability under Thermal Action. Journal of Performance of Constructed Facilities. 32. 10.1061/(ASCE)CF.1943-5509.0001212.

-Yang, Dong-Hui & Yi, Ting-Hua & Li, Hong-Nan & Zhang, Yu-Feng. (2018). Monitoring and analysis of thermal effect on tower displacement in cable-stayed bridge. Measurement, Volume 115, 2018.

Point 3:

Lines 431-432: Would it have a basis for the sampling rate of 10 Hz? Whether the value of the sampling rate should be changed for different practical bridges? If so, could you describe a general method to determine the value of the sampling rate, or it necessarily needs to be selected subjectively? Please explain it.

Response 3: It has been added these lines at the end of the discussion section, to clarify how to determine the sampling rate, we hope:

A limitation or disadvantage of the TDCP algorithm is the requirement of a high sampling rate of the receivers. Operating at 10 Hz is essential for the success of the method. In this way, the errors in such close epochs are very similar and, by subtracting carrier phases, can be eliminated. Geodetic receivers can record data at 10 Hz or even more in some cases.

A sampling rate of 1 Hz was not enough for the bridge case because the standard error was around 0.09 m (obtained from expression (17)), and the size of the displacements was very similar.

The sampling rate is correlated with the level of accuracy needed in the project. In our case, the level of accuracy selected was 0.01 m. Expression (17) allows knowing the standard deviations of the TDCP method. The standard deviations will be compared to the level of accuracy needed in each project.

Nowadays it is not possible to use mass-market GNSS devices with TDCP technique because their sampling rate is lower than 10 Hz. However, the current trend is to use low-cost GNSS sensors, especially with the Internet of Things (IoT). IoT allows permanent control of the structures, 7 days 24 hours, to detect anomalies and to have a precise model of the bridge movements. Being that a superior quantity of data will improve the accuracy of the mean measurements.

Point 4:

Line 512: The “RTK” does not appear in the previous sections. Please explain its meaning here and revise the manuscript.

Response 4: It has been added these lines at the end of Section 3, Experiments and results:

Real-Time Kinematic (RTK) is a positioning technique that uses carrier phase and pseudorange measurements. The position calculations are performed using the base station and the rover. In this study, R1, R2, and R3 receivers are the rovers. The data observation with this methodology requires the base station, which is working simultaneously with the rover receivers and is located outside the bridge. RTK is used for applications that require higher accuracies, such as centimeter-level positioning, up to 1 cm + 1 ppm.

The professional software Leica Infinity was used to obtain a post-processed RTK solution of the relative displacements of the bridge and associated standard deviation in ENU local system.

The RTK method is lower accurate than the TDCP, but we could not find a better kinematic solution to compare to the TDCP solution. Tables 2 and 3 show the means, as averages or arithmetic means, of the all-standard deviations associated with every position for a 30-minute period. The TDCP solution has a standard deviation means of around 0.005 m and the RTK solution around 0.015 m.

The maximum displacements obtained from the TDCP technique were evaluated against the maximum displacements of the post-processed RTK solution. The maximum displacements obtained from the TDCP technique were evaluated against the maximum displacements of the post-processed RTK, as a well-known method. The results of both methods are very similar. The largest displacement of the bridge is de vertical direction (see ∆U-TDCP values in Table 2 and ∆U-RTK values in Table 3). The ∆U-TDCP values for R1, R2, and R3 receptors were: 0.196 m, 0.190 m, and 0.0149 m respectively. The ∆U-RTK values for R1, R2, and R3 receptors were: 0.198 m, 0.149 m, and 0.018 m respectively. So, the TDCP solution is in accordance with the RTK solution.

 Point 5:

Lines 576 and 592: Please check and correct the errors in the manuscript.

Response 5: We have removed the unintended references.

 Point 6:

 Figures 6-9: The conclusion that " Recover the initial position after a movement " is not clear and obvious in Figures 6-9. Please describe it in detail and revise the manuscript if considered necessary.

 Response 6: Following previous investigations by the authors [Real-time high-rise building monitoring system using global navigation satellite system technology. Measurement 123 (2018) https://doi.org/ 10.1016/j.measurement.2018.03.05], observational data were taken over the three-hour period at night to define the initial position of the Assut de l'Or. During this period traffic was practically non-existent on the bridge, as well as the traffic load, with no movement of it.

The graphs have their origin at the initial position defined for the bridge, and during the observation period, the graphs oscillate about that origin. This determines that the bridge recovers its position after the excitation produced by the traffic load.

The last paragraph added in section 3, to improve the understanding is this:

Recovering the initial position after movement is a key factor in the maintenance and detection of the potential structural problems of a bridge. To detect that recovery, we have obtained the means of all displacements in a 30-minute period, using the TDCP solution. If the means, as averages, are close to 0 m, the recoverable deformation in the period of time could be evidenced. Table 4 presents the means of the east, north, and vertical displacements, all of them lower than 0.004 m. It could be said that most of the time the bridge remains in the zero position.

Point 7:

Tables 2 and 4: What is the basis for evaluating the RTK and TDCP methods based on the standard deviation?

Response 7:

-Part of this point is answered in point 4, we think.

The standard deviation is obtained to compare the accuracy of both positioning methods: RTK and TDCP. RTK has lower accuracy than TDCP.

The maximum displacements obtained from the TDCP technique were evaluated against the maximum displacements of the post-processed RTK, as a well-known method. The results of both methods are very similar.

-From Table 2 we have changed and added new lines, in order to explain and clarify the confusing text. We hope we have improved the concepts.

Point 8:

Vehicle-induced displacement response is vital for bridge monitoring. Could you describe whether the proposed method can accurately capture the vehicle excitation and measure the vehicle-induced displacement? If so, what level of measurement accuracy can be achieved?

 Response 8: The Assut de l'Or bridge is officially open to traffic in December 2008. During the current study, the bridge is permanently open to road traffic, car and bicycle, and pedestrian traffic.

The traffic flow data was provided by the traffic regulation section of the Valencia City Council. In this way, the traffic flow data can be temporarily correlated with the displacements measured on the bridge. This situation made it possible to evaluate the response to dynamic excitation of the bridge deck as a function of daily traffic.

And about the level of measurement accuracy, the TDCP solution has a standard deviation means of around 0.005 m. Standard deviations are shown in Table 2.

Round 2

Reviewer 2 Report

Thank you for addressing all my comments. I do not have any additional comments for you.